# Optimization-Based Resource Management Algorithms with Considerations of Client Satisfaction and High Availability in Elastic 5G Network Slices

**DOI:** 10.3390/s21051882

**Published:** 2021-03-08

**Authors:** Chiu-Han Hsiao, Frank Yeong-Sung Lin, Evana Szu-Han Fang, Yu-Fang Chen, Yean-Fu Wen, Yennun Huang, Yang-Che Su, Ya-Syuan Wu, Hsin-Yi Kuo

**Affiliations:** 1Research Center for Information Technology Innovation, Academia Sinica, Taipei 115, Taiwan; yennunhuang@citi.sinica.edu.tw (Y.H.); R06725052@ntu.edu.tw (Y.-C.S.); R05725060@ntu.edu.tw (H.-Y.K.); 2Department of Information Management, National Taiwan University, Taipei 10617, Taiwan; flin@ntu.edu.tw (F.Y.-S.L.); D06725003@ntu.edu.tw (E.S.-H.F.); D09725003@ntu.edu.tw (Y.-F.C.); R06725016@ntu.edu.tw (Y.-S.W.); 3Graduate Institute of Information Management, National Taipei University, New Taipei City 23799, Taiwan; yeanfu@mail.ntpu.edu.tw

**Keywords:** network slicing, resource allocation, load balancing, admission control, high availability, Lagrangian relaxation (LR)

## Abstract

A combined edge and core cloud computing environment is a novel solution in 5G network slices. The clients’ high availability requirement is a challenge because it limits the possible admission control in front of the edge cloud. This work proposes an orchestrator with a mathematical programming model in a global viewpoint to solve resource management problems and satisfying the clients’ high availability requirements. The proposed Lagrangian relaxation-based approach is adopted to solve the problems at a near-optimal level for increasing the system revenue. A promising and straightforward resource management approach and several experimental cases are used to evaluate the efficiency and effectiveness. Preliminary results are presented as performance evaluations to verify the proposed approach’s suitability for edge and core cloud computing environments. The proposed orchestrator significantly enables the network slicing services and efficiently enhances the clients’ satisfaction of high availability.

## 1. Introduction

Novel applications of 5G services are cataloged with three kinds of quality of service (QoS) features: massive machine-type communication (mMTC), enhanced mobile broadband (eMBB), and ultra-reliable low-latency communication (uRLLC) in 5G cellular networks [1]. Two system architectures have been proposed for ensuring the scalability and flexibility of resource allocation and scheduling for the diverse QoS requirements. The first highly acceptable approach is the cloud radio access network (C-RAN). The baseband processing and networking functions are virtually centralized in a resource pool for resource scalability and flexibility [2]. The other one is mobile edge computing (MEC). It supports interactive and real-time applications associated with nearby cloud hosts to obtain the required latency for addressing urgent and distributed requirements [3]. To implement C-RAN and MEC, software-defined networking (SDN) and network function virtualization (NFV) have been integrated to develop network slicing technologies in 5G [4]. Network slices are end-to-end (E2E) mutually separate sets of programmable infrastructure resources with independent control. A slice is a logical network adapted for particular applications regarding the diverse QoS requirements [1].

However, SDN and NFV have paved the way for employing the slicing concept. The network slice-as-a-service (NSaaS) strategy has several problems and challenges [4,5]. Furthermore, the failure of virtualized network functions (VNFs) may impact the QoS for service provisioning of the control plane (e.g., MME) and the data plane (e.g., S-GWsor PDN-GWs), respectively [4]. The related factors would be addressed and collected to design an orchestrator with the resource management efficiently and effectively for the VNFs, such as resource allocation, load balancing, and high availability (HA) for building up a high performance and robust slicing network.

Furthermore, HA is considered a high stringency requirement for providing a sustainable business with emergency response applications to support a reliable performance with zero downtime and costs. Some technologies can be adopted by the redundant array of independent discs, the replication of nodes, or master-slave database redundancy to offer data protection in an HA-equipped system [6,7,8]. From the network operator perspective, 5G network slices’ resources are limited for simultaneous profit maximization. Load balancing is typically adopted by an orchestrator to achieve resource planning and provisioning and enhance performance [4,9].

In this paper, the efficient elastic mechanisms of resource management in virtual machines (VMs) with VNFs are proposed to integrate admission control, load balancing, resource allocation, and HA arrangement in 5G network slices. The system architecture is shown in Figure 1. An integer programming problem is formulated to maximize system utilization, subject to the quality of services, resource requirements, and HA constraints. The resource allocation problem is combined with knapsack and bin-packing problems. Generally, the VNFs or called tasks with various amounts of resource requirements emulated as VMs should be packed into a finite number of servers. A few servers in active status and then stepping forward servers by increasing demands are possible to reduce the operating costs in a bin-packing problem. A set of tasks for applications with diverse weights and values is selected for inclusion to remain within the resource limit and reach the maximum benefit, a well-known knapsack problem. Furthermore, the knapsack problem is the aforementioned combinatorial nondeterministic polynomial time (NP) hard problem. One possible solution is adopted by the Lagrangian relaxation (LR) method efficiently and effectively. Near-optimal solutions provide practical methods to overcome the bottleneck within limited computing resources of slices. The virtualization, parallel computing, and mathematical programming techniques are also developed in this paper to make near-optimal decisions from an NSaaS service provider’s perspective. The proposed resource management algorithms ensure that computing resources are equitably distributed in the slices of cloudlets and core clouds. The computation in the edge and core clouds is performed to achieve maximum resource utilization with HA and minimal cost. Moreover, the admission control scheme with the additional HA requirements is devised to admit the maximum number of jobs. Each job requires various types of resources.

The remainder of this paper is organized as follows: Section 2 reviews related works. Section 3 introduces the mathematical model and problem description. The LR-based solution approach is developed, as shown in Section 4. Section 5 presents computational experiments. Conclusions are drawn in Section 6.

## 2. Literature Review

A cloud computing environment in 5G network slices supports share-based services in a pay-as-you-go approach. It provides operators with a flexible architecture from the perspective of a network operator. Resource management with complex on-demand traffic in the share-based cloud computing environment is a considerable challenge. Inappropriately designed resource management solutions might lead to the network’s increasing costs. This is related to unsatisfactory quality of services (QoS) and system reliability regarding the penalty of user experience and call-blocking probability due to the virtualized network function failure. Table 1, Table 2 and Table 3 compare the resource management problems investigated with those investigated in related studies in terms of (i) resource allocation, (ii) load balancing, and (iii) admission control.

### 2.1. Resource Allocation

For the maximization of the benefits of the cloud system and service quality enhancement, Liu et al. [10] developed a joint multi-resource allocation model based on a semi-Markov decision process (SMDP). They solved the optimization problem using linear programming to attain wireless resource allocation and near-optimal decisions in cloud computing. However, cloud service providers fulfill individual requirements to achieve a win-win situation for both parties in terms of computational efficiency, budget balance, and truthfulness. Jin et al. [11] designed an incentive-compatible auction mechanism to appropriately allocate matching among cloud resources and user demands. In this auction model, the buyers are mobile devices, and the sellers are cloudlets. The nearest control center can adopt the role of the auctioneer to reduce transmission costs and latency. Furthermore, by assuming that all cloudlets can provide the same resources with distinct rewards, Liu and Fan [12] presented a two-stage optimization strategy to achieve optimal cloudlet selection in a multi-cloudlet environment and optimal resource allocation in response to the request in the cloudlet. Studies on resource allocation in a cloud computing environment are shown in Table 1. Resource allocation and scheduling algorithms have been proposed in numerous research areas, such as transportation management, industrial management, operational research, computer science, and particularly in real-time operating systems [13,14]. For instance, the earliest deadline first (EDF) is a dynamic scheduling algorithm used in real-time operating systems to allocate computing resources in central processing units (CPUs) using a priority queue. The queue searches for the task with the closest deadline; if a job cannot be completed within its time frame, the operating system must release the job. The main idea is to maximize resource use for several competing interests while balancing the load. A weighted priority for each data flow can be assigned inversely proportional to the respective flow’s anticipated resource consumption [15,16]. Ding et al. [17] proposed a Linux scheduling policy with a priority queue rather than a first-in-first-out (FIFO) queue to improve kernel-based virtual machine performance. Zhao et al. [18] proposed online VM placement algorithms to cost efficiently allocate resources to VMs to increase revenues in a managed server farm. First-fit (FF), FF migration (FFM), least reliable first (LRF), and decreased density greedy (DDG) algorithms are packing strategies relevant to task optimization for achieving desirable performance.

**Table 1 sensors-21-01882-t001:** Resource allocation comparisons with existing methods. SMDP, semi-Markov decision process; EDF, earliest deadline first; FF, First-fit; FFM, FF migration; LRF, least reliable first; DDG, decreased density greedy.

Classification	Objective	Strategy	Related Studies	Proposed Methods
Resource allocation	Maximizing revenue and satisfying QoS	SMDP; auction; EDF; FIFO; FF; FFM; LRF; DDG	[10,11,12,13,14,15,16,17,18]	LR and next-fit are adopted
**Comparison with related studies**				
This paper regards offloading tasks as virtual machines with distinct rewards with various levels of demands in the cloudlet and core cloud environment. The Lagrangian relaxation-based approach is proposed to maximize system revenue, subject to constraints such as computing capacity, assignments, and quality of service requirements. The objective is to maximize the total value of tasks by using proposed heuristics to appropriately rearrange resources using the next-fit algorithm to allocate tasks and satisfy application requirements.				

Based on the analysis of the previous studies, this paper regards offloading tasks as virtual machines with distinct rewards with various levels of demands in the cloudlets and core clouds. The Lagrangian relaxation-based approach is proposed to maximize system revenue, subject to constraints such as computing capacity, assignments, and quality of service requirements. The approach is adopted by considering finding the optimal solutions other than SMDP within limited states or constraints. The objective is to maximize the total value of tasks using proposed heuristics in polynomial time. It also appropriately rearranges resources using the next-fit algorithm to allocate tasks and satisfy application requirements.

### 2.2. Load Balancing

Load balancing is a crucial technology that optimizes mobile application performance. Several studies have proposed cloudlet technology solutions [19] to achieve load balancing for mobile devices (Table 2). Jia et al. [19] devised a load-balancing algorithm to balance the workload among multiple cloudlets in wireless metropolitan area networks. A redirection scheme shortened the average offload task response time and improved user experience. Yao et al. [20] studied load balancing for cloudlet nodes. The task allocation problem was formulated as an integer linear problem. A two-step centralized appointment-driven strategy was used to obtain a solution with minimal average response time and a balanced load. From a network operator perspective, time-varying and periodically changing demands are challenges in providing on-demand services. Furthermore, cloud service providers face load-balancing problems in cloud computing environments. The system availability of on-demand access should be considered using an adjustable resource assignment mechanism to satisfy QoS. If resource demands are stochastic, their accurate forecast is severe. Historical data are measured for estimating usage data to determine the load distribution and evaluate the proposed algorithm [21].

**Table 2 sensors-21-01882-t002:** Load balancing comparisons with existing methods.

Classification	Objective	Strategy	Related Studies	Proposed Methods
Load balance	Maximum system availability	Bin packing; 0/1 knapsack	[19,20,21]	Based on the system utilization in global view
**Comparison with related studies**				
In general, as many tasks as possible are admitted to maximize total revenue; however, the supply and demand for resources are unbalanced during peak traffic hours. This study determines which tasks are selected and dispatched to achieve maximum objective values subject to task assignment and limited capacity. The problems are classified as bin-packing and 0/1 knapsack problems. Herein, the assignment decisions are based on resource utilization functions, including a central processing unit, random access memory, hard drive, and bandwidth for each server type. The assignment strategies are designed to fit the QoS requirements, such as those of computing and transmission in variant traffic loads in a global view adopted by the heuristics of bin-packing strategies.				

In general, as many tasks as possible are admitted to maximize total revenue; however, the supply and demand for resources are unbalanced during peak traffic hours. This study determines the orchestrator’s decisions for which tasks are selected and dispatched to achieve the maximum objective values subject to task assignment and limited capacity. The problems are classified as the bin-packing and 0/1 knapsack combined problems. Herein, the orchestrator’s decisions are based on resource utilization functions, including a central processing unit, random access memory, hard drive, and bandwidth for each server type. The orchestrator’s objective is also designed for that assignment strategy fitting the QoS requirements, such as those of computing and transmission in variant traffic loads in a global view adopted by the heuristics of bin-packing strategies.

### 2.3. Admission Control

Preventing server overload and ensuring application performance are the goals of admission control [22]. This mechanism decides whether to admit a particular service request to the server. It is implemented for meeting QoS requirements and achieving service providers’ expected revenue [23]. Hoang et al. [24] developed a semi-Markov decision process (SMDP)-based optimization model for mobile cloud computing that considers constraints such as resources, bandwidth, and QoS to perform admission control tasks. This framework ensures QoS performance and maximizes the reward. Xia et al. [25] aimed to optimize cloud system throughput. An effective admission algorithm based on the proposed resource cost paradigm to model various resource consumptions was devised according to the online request following the current workload. Table 3 presents the comparison of the admission control methods.

**Table 3 sensors-21-01882-t003:** Admission control comparisons with existing methods.

Classification	Objective	Strategy	Related Studies	Proposed Methods
Admission control	Maximum QoS and throughput	SMDP; priority; no-priority; 0/1 knapsack	[22,23,24,25]	Non-priority and Lagrangian multipliers are adopted
**Comparison with related studies**				
The semi-Markov decision process (SMDP) approach relies on partial data, such as resources, bandwidth, and quality of service (QoS), to decide whether to accept or reject a task. SMDP computing time is inefficient for jobs with time constraints, such as delay and delay tolerance sensitive applications. The priority of user information is unknown for decision making under a computing time constraint. In this paper, call admission control mechanisms within the conditions with non-priority and QoS constraints are jointly considered. The Lagrangian relaxation-based approach is proposed to maximize system revenue combined with the proposed resource allocation methods to admit tasks and satisfy application requirements appropriately.				

The SMDP-based approach relies on partial data. The computing time is also inefficient for jobs with time constraints, such as delay and delay tolerance sensitive applications. The priority of user information is unknown for decision making under a computing time constraint. In this paper, call admission control mechanisms within the conditions with non-priority and QoS constraints are jointly considered. The Lagrangian relaxation-based approach is proposed to maximize system revenue combined with the proposed resource allocation methods to appropriately admit tasks and satisfy application requirements.

### 2.4. Research Scope

This research focuses on resource management in various scenarios in 5G network slices. An optimization-based approach (LR) is used to solve the mathematical programming problem to maximize system revenue. In our proposed cloud computing framework (cloudlets and core clouds) in 5G network slices, two primary research questions are considered. What is the most efficient algorithm for resource allocation within admission control, resource scheduling, and load-balancing policies in a limited-resource cloud computing environment? Is the HA of VMs considerably influenced in the system?

Additionally, the problem is addressed under rapidly increasing data traffic conditions and a limited resource pool to obtain near-optimal policies using a combination of LR approaches. The solutions satisfy the QoS-related constraints in transmission and stand on computation perspectives to fulfill the throughput, delay, and delay jitter requirements. They are compared with other resource management schemes for efficiency and effectiveness.

## 3. Mathematical Formulation

A mathematical model is proposed to manage an optimization programming problem. It focuses on developing a well-designed algorithm in a parallel computing environment for admission control, resource scheduling, and an HA arrangement to maximize cloud service provider profits in 5G network slices. Based on the combined cloudlet and core cloud network architecture (Figure 1), VMs with VNFs are requested by various applications and require specific resources. They should be allocated to the server *k* with computation and transmission capacities concerning CPU, RAM, storage, and internal or shared bandwidth, expressed as $P_{k}$, $M_{k}$, $H_{k}$, $B{}_{k}^{n}$, and $B^{c}$, respectively, in cloudlets and core clouds. The proposed model in the orchestrator supports network resource management in front of the cloudlet and core cloud environments when assuming that resource management strategies are followed when facing a batch of requests. Table 4 and Table 5 present lists of the given parameters and decision variables.

The system’s profit is the combination of the rewards of admitting applications, the penalties of rejecting the other applications, and the cost for turning on the servers.

The objective function $Z_{\mathrm{IP}}$ is shown in Equation (Equation 1), and its goal is to maximize system profits among all VMs requested by the applications.
(1)$$Z_{\mathrm{IP}}=\sum_{i\in I}R_{i}y_{i}-\sum_{i\in I}P_{i}\left(1-y_{i}\right)-\sum_{s\in S}\beta_{s}z_{s}$$

To acquire the highest revenue, the optimization problem is shown as:$$\begin{array}{cc} \; & \mathrm{Objective}\;\mathrm{function}\text{:}\\ \; & \max\;Z_{\mathrm{IP}}=\min-Z_{\mathrm{IP}}\\ \; & \mathrm{subject}\;\mathrm{to}\text{:}\;\\ \; & C1\text{:}\;(|W_{i}\left|+\right|T_{i}\left|\right)y_{i}\leq \sum_{j\in W_{i}}\sum_{s\in S}a_{ijs}+\sum_{\ell \in T_{i}}\sum_{s\in S}b_{i\ell s},\;\forall i\in I,\\ \; & C2\text{:}\;\sum_{s\in S}a_{ijs}\leq 1,\;\forall i\in I,\;\forall j\in W_{i},\\ \; & C3\text{:}\;\sum_{s\in S}b_{i\ell s}\leq 1,\;\forall i\in I,\;\forall \ell \in T_{i},\\ \; & C4\text{:}\;\sum_{i\in I}\sum_{j\in W_{i}}a_{ijs}+\sum_{i\in I}\sum_{\ell \in T_{i}}b_{i\ell s}\leq V_{s},\;\forall s\in S,\\ \; & C5\text{:}\;\sum_{i\in I}\sum_{j\in W_{i}}D_{ij}a_{ijs}+\sum_{i\in I}\sum_{\ell \in T_{i}}D_{i\ell }b_{i\ell s}\leq E_{s},\;\forall s\in S,\\ \; & C6\text{:}\;\sum_{i\in I}\sum_{j\in W_{i}}G_{ij}a_{ijs}+\sum_{i\in I}\sum_{\ell \in T_{i}}G_{i\ell }b_{i\ell s}\leq M_{s},\;\forall s\in S,\\ \; & C7\text{:}\;\sum_{i\in I}\sum_{j\in W_{i}}Q_{ij}a_{ijs}+\sum_{i\in I}\sum_{\ell \in T_{i}}Q_{i\ell }b_{i\ell s}\leq H_{s},\;\forall s\in S,\\ \; & C8\text{:}\;\sum_{i\in I}\sum_{j\in W_{i}}\sum_{\ell \in T_{i}}X_{ij\ell }f_{ij\ell s}\leq B_{s}^{n},\;\forall s\in S,\\ \; & C9\text{:}\;\sum_{i\in I}\sum_{j\in W_{i}}\sum_{\ell \in T_{i}}X_{ij\ell }\left(1-f_{ij\ell s}\right)\leq B^{c},\;\forall s\in S,\\ \; & C10\text{:}\;a_{ijs}\leq z_{s},\;\forall i\in I,\;\forall j\in W_{i},\;\forall s\in S,\\ \; & C11\text{:}\;b_{i\ell s}\leq z_{s},\;\forall i\in I,\;\forall \ell \in T_{i},\;\forall s\in S,\\ \; & C12\text{:}\;a_{ijs}+b_{i\ell s}\leq f_{ij\ell s}+1,\;\forall i\in I,\;\forall j\in W_{i},\;\forall \ell \in T_{i},\;\forall s\in S. \end{array}$$
where C1–C3 belong to the admission control and assignment constraints, C4–C9 belong to the capacity constraints, and C10–C12 belong to the HA constraints.

For admission control and assignment constraints, $\left(|W_{i}|+|T_{i}|\right)$ is the total number of standard or additional HA VMs required by an application, in which the VMs are in disjoint sets, where $W_{i}⋂T_{i}=⌀$ in constraint C1. $\sum_{s\in S}a_{ijs}$ is the number of standard VMs that must be admitted and allocated into servers. $\sum_{s\in S}b_{i\ell s}$ is the number of admitted and allocated HA VMs. The total number of allocated HA and standard VMs should be greater than or equal to the requirement, as shown in constraint C1. That is, while application *i* is completely served, the summation of $a_{ijs}$ and $b_{i\ell s}$ should be greater than and equal to the demand on the right-hand side of constraint C1.

$\sum_{s\in S}a_{ijs}$ and $\sum_{s\in S}b_{i\ell s}$ are shown in the constraints C2 and C3, which means when the value of $\sum_{s\in S}a_{ijs}$ or $\sum_{s\in S}b_{i\ell s}$ is less than or equal to one, the VMs are inseparably assigned to servers. In other words, a virtual machine is not partially allocated to servers.

For capacity constraints, the resources offered by a server are defined as a set of four factors: the maximum number of VMs in the server *s* ($V_{s}$), the processing capacity of each CPU core ($E_{s}$), the RAM capacity ($M_{s}$), and the storage capacity ($H_{s}$). The total resources required by VMs for each server cannot exceed its available resources as formulated in the constraints C4–C7.

We also set the internal bandwidth rate ($B_{s}^{n}$) in the server *s* for internal transmission between VMs. For example, two applications, *i* and i+1, and their index sets of VMs are $W_{i}=\left\{1_{i},2_{i}\right\}$, $T_{i}=\left\{3_{i}\right\}$, $W_{i+1}=\left\{1_{i+1},2_{i+1}\right\}$, and $T_{i+1}=\left\{3_{i+1}\right\}$, as shown in Figure 2. The link between VM $3_{i}$ and VM $2_{i+1}$ represents the internal bandwidth required by VM $3_{i}$ to connect to VM $2_{i+1}$ in server *s*. The constraint C8 indicates that the total bandwidth required by all the VMs should not exceed the internal bandwidth $B_{s}^{n}$ in server *s*. The external bandwidth rate ($B^{c}$) is set for the transmission between servers in the cloud computing environment (cloudlets and core clouds), as illustrated in Figure 2. The link between server s−1 and server *s* represents the bandwidth requested by VM $1_{i}$ to connect to VM $3_{i}$. The constraint C9 indicates that the total bandwidth required by all VMs accepted by servers in a cloud should not exceed the external bandwidth.

In this work, we assume two kinds of VMs, standard VMs and VMs with HA, of the same application, cannot be assigned to the same server, as shown in Figure 2. In the beginning, all servers contain no VMs; then, for example: application *i* comes in and needs two standard VMs and one VM with HA, as mentioned in the previous paragraph. First, we put two standard VMs, VM $1_{i}$ and VM $2_{i}$, in server s−1, while the two kinds of VMs cannot be assigned to the same server, then VM $3_{i}$ has to be assigned to server *s*. To continue, the next application i+1 comes in, and we put its standard VMs $1_{i+1}$ and $2_{i+1}$ in server s−1 and server *s*, respectively, while server s−1 and server *s* have no residual capacity to handle one more VM. Meanwhile, the VM with HA, VM $3_{i+1}$, cannot be put in server s−1 or server *s* due to our exclusive assumption. Thus, VM $3_{i+1}$ is assigned to server s+1.

For HA configuration constraints, the decision variable $f_{ij\ell s}$ for application *i* is set for the VM assignments. The index sets *j* and *ℓ* for different kinds of VMs must be collocated to server *s* and separated into different servers, where $j\in W_{i}$ and $\ell \in T_{i}$. The relationships are expressed as the constraint C12. The constraints C10 and C11 indicate the server power status. If any VM is admitted and assigned to server *s*, server *s* must be powered on, and $z_{s}$ should be set to one. The constraint C12 assures that when VMs *j* and *ℓ* are assigned to the same server *s*, which means that $a_{ijs}$ and $b_{i\ell s}$ are both set to one, the exclusive setting $f_{ij\ell s}$ must also be one. In other words, the standard VM and the HA VM requested from the same application cannot be allocated to the same server *s*.

## 4. Lagrangian Relaxation-Based Solution Processes

The Lagrangian relaxation method is proposed for solving large-scale mathematical programming problems, including optimization problems with linear, integer, and nonlinear programming problems in many practical applications [26]. The key idea is to relax complicated constraints into a primal optimization problem and extend feasible solution regions to simplify the primal problem. Based on the relaxation, the primal problem is transformed into an LR problem associated with Lagrangian multipliers [27,28,29]. Figure 3 illustrates the six procedural steps in the LR method.

### 4.1. Procedures of Step 1: Relaxation

To separate the feasible region of the primal problem into several subproblems with an independent set of decision variables, the primal problem is transformed into an LR problem. The relaxation method associated with Lagrangian multipliers is applied to the constraints C1 and C4–C13 in Step 1, as presented in Figure 3. Then, the constraints C1 and C4–C12 with the multipliers are added to the primal problem Equation (Equation 1), as shown in Equation (Equation 2) and denoted as $Z_{\mathrm{LR}}$.
(2)$$\begin{array}{cccc} \; & Z_{\mathrm{LR}} & = & -\sum_{i\in I}R_{i}y_{i}+\sum_{i\in I}P_{i}\left(1-y_{i}\right)+\sum_{s\in S}\beta_{s}z_{s}\\ \; & \; & & +\sum_{i\in I}\mu_{i}^{1}\left[\left(\right|W_{i}\left|+\right|T_{i}\left|\right)y_{i}-\sum_{j\in W_{i}}\sum_{s\in S}a_{ijs}-\sum_{\ell \in T_{i}}\sum_{s\in S}b_{i\ell s}\right]\\ \; & \; & & +\sum_{s\in S}\mu_{s}^{2}\left[\sum_{i\in I}\sum_{j\in W_{i}}a_{ijs}+\sum_{i\in I}\sum_{\ell \in T_{i}}b_{i\ell s}-V_{s}\right]\\ \; & \; & & +\sum_{s\in S}\mu_{s}^{3}\left[\sum_{i\in I}\sum_{j\in W_{i}}D_{ij}a_{ijs}+\sum_{i\in I}\sum_{\ell \in T_{i}}D_{i\ell }b_{i\ell s}-E_{s}\right]\\ \; & \; & & +\sum_{s\in S}\mu_{s}^{4}\left[\sum_{i\in I}\sum_{j\in W_{i}}G_{ij}a_{ijs}+\sum_{i\in I}\sum_{\ell \in T_{i}}G_{i\ell }b_{i\ell s}-M_{s}\right]\\ \; & \; & & +\sum_{s\in S}\mu_{s}^{5}\left[\sum_{i\in I}\sum_{j\in W_{i}}Q_{ij}a_{ijs}+\sum_{i\in I}\sum_{\ell \in T_{i}}Q_{i\ell }b_{i\ell s}-H_{s}\right]\\ \; & \; & & +\sum_{s\in S}\mu_{s}^{6}\left[\sum_{i\in I}\sum_{j\in W_{i}}\sum_{\ell \in T_{i}}X_{ij\ell }f_{ij\ell s}-B_{s}^{n}\right]\\ \; & \; & & +\sum_{s\in S}\mu_{s}^{7}\left[\sum_{i\in I}\sum_{j\in W_{i}}\sum_{\ell \in T_{i}}X_{ij\ell }\left(1-f_{ij\ell s}\right)-B^{c}\right]\\ \; & \; & & +\sum_{i\in I}\sum_{j\in W_{i}}\sum_{s\in S}\mu_{ijs}^{8}\left[a_{ijs}-z_{s}\right]\\ \; & \; & & +\sum_{i\in I}\sum_{\ell \in T_{i}}\sum_{s\in S}\mu_{i\ell s}^{9}\left[b_{i\ell s}-z_{s}\right]\\ \; & \; & & +\sum_{i\in I}\sum_{j\in W_{i}}\sum_{\ell \in T_{i}}\sum_{s\in S}\mu_{ij\ell s}^{10}\left[a_{ijs}+b_{i\ell s}-f_{ij\ell s}-1\right] \end{array}$$

Then, the optimization problem can be reformulated as:$$\begin{array}{cc} \; & \mathrm{Objective}\;\mathrm{function}\text{:}\;\\ \; & \min\;Z_{\mathrm{LR}}\\ \; & \mathrm{subject}\;\mathrm{to}\text{:}\;C2,\;C3, \end{array}$$
where $y_{i}\in \left\{0,1\right\}$, $a_{ijs}\in \left\{0,1\right\}$, $b_{i\ell s}\in \left\{0,1\right\}$, $f_{ij\ell s}\in \left\{0,1\right\}$, and $z_{s}\in \left\{0,1\right\}$.

### 4.2. Procedures of Steps 2 and 3: Decomposition and Solving Subproblems

The LR problem can be decomposed into several independent subproblems, with their related decision variables. The divide-and-conquer approach is used to solve the subproblems correspondingly.

#### 4.2.1. Subproblem 1 (Related to $y_{i}$)

By extracting items with decision variable $y_{i}$, the optimization problem of Subproblem 1 can be developed as:(3)$$\begin{array}{cc} \; & \mathrm{Objective}\;\mathrm{function}\text{:}\;\\ \; & \min\;\sum_{i\in I}\left(-R_{i}-P_{i}+\mu_{i}^{1}|W_{i}|+\mu_{i}^{1}|T_{i}|\right)y_{i}\\ \; & \mathrm{subject}\;\mathrm{to}\text{:}\;y_{i}\in \left\{0,1\right\}. \end{array}$$

Equation (Equation 3) can be divided into |I| independent subproblems. For each application *i*, where i∈I, the decision variable $y_{i}$ is set to one when the coefficient $\left(-R_{i}-P_{i}+\mu_{i}^{1}|W_{i}|+\mu_{i}^{1}|T_{i}|\right)$ is less than zero. Otherwise, $y_{i}$ is set to zero. The run time is O(|I|), and the pseudocode is illustrated in Algorithm 1.
**Algorithm 1:** Subproblem 1.**Input:** Given parameters R, P, W, T and Lagrangian    multipliers $\mu^{1}$.**Output:** Decision variable y.**Initialize:**$y_{i}\leftarrow 0$, ∀i∈I**for**i=0 to (|I|−1)
**do**    $c\leftarrow -R_{i}-P_{i}+\mu_{i}^{1}|W_{i}|+\mu_{i}^{1}|T_{i}|$    **if**
c<0
**then**        $y_{i}\leftarrow 1$    **end if****end for**

#### 4.2.2. Subproblem 2 (Related to $a_{ijs}$)

By extracting items with decision variable $a_{ijs}$, the optimization problem of Subproblem 2 can be developed as:(4)$$\begin{array}{cc} \; & \mathrm{Objective}\;\mathrm{function}\text{:}\;\\ \; & \min\;\sum_{i\in I}\sum_{j\in W_{i}}\sum_{s\in S}(-\mu_{i}^{1}+\mu_{s}^{2}+\mu_{s}^{3}D_{ij}+\mu_{s}^{4}G_{ij}+\mu_{s}^{5}Q_{ij}\\ \; & \;\;\;\;\;\;\;\;\;\;\;\;\;\;\;\;\;+\mu_{ijs}^{8}+\sum_{\ell \in T_{i}}\mu_{ij\ell s}^{10})a_{ijs}\\ \; & \mathrm{subject}\;\mathrm{to}\;C2\text{:}\;\sum_{s\in S}a_{ijs}\leq 1,\;\forall i\in I,\;\forall j\in W_{i},\\ \; & \;\;\;\;\;\;\;\;\;\;\;\;\;\;\;\;\;a_{ijs}\in \left\{0,1\right\}. \end{array}$$

Equation (Equation 4) can be divided into $\left|I|\right|W_{i}\left||S\right|$ cases. The decision variable $a_{ijs}$ is set to one, and the minimum coefficient $\left(-\mu_{i}^{1}+\mu_{s}^{2}+\mu_{s}^{3}D_{ij}+\mu_{s}^{4}G_{ij}+\mu_{s}^{5}Q_{ij}+\mu_{ijs}^{8}+\sum_{\ell \in T_{i}}\mu_{ij\ell s}^{10}\right)$ is less than zero and corresponds to alliance subindices *i*, *j*, and *s*. To satisfy the constraint C2, the decision variable should be set to one only if the minimum coefficient with subindex *s* is a required sorting processes. Otherwise, $a_{ijs}$ is set to zero. The run time is $O(|I||W_{i}||S||T_{i}|)$. The pseudocode is illustrated in Algorithm 2.
**Algorithm 2:** Subproblem 2.**Input:** Given parameters D,G,Q and Lagrangian    multipliers $\mu^{1},\mu^{2},\mu^{3},\mu^{4},\mu^{5},\mu^{8},\mu^{10}$.**Output:** Decision variable a.**Initialize:**$a_{ijs}\leftarrow 0$, $\forall i\in I,\forall j\in W_{i},\forall s\in S$**for**i=0 to (|I|−1)
**do**    **for**
j=0 to $\left(|W_{i}|-1\right)$
**do**        **for**
s=0 to (|S|−1)
**do**           $c_{s}\leftarrow -\mu_{i}^{1}+\mu_{s}^{2}+\mu_{s}^{3}D_{ij}+\mu_{s}^{4}G_{ij}+\mu_{s}^{5}Q_{ij}$                  $+\mu_{ijs}^{8}+\sum_{\ell \in T_{i}}\mu_{ij\ell s}^{10}$        **end for**        Find the index *m* that $c_{m}$ has the minimum value        in c.        **if**
$c_{m}<0$
**then**           $a_{ijm}\leftarrow 1$        **end if**    **end for****end for**

#### 4.2.3. Subproblem 3 (Related to $b_{i\ell s}$)

By extracting items with decision variable $b_{i\ell s}$, the optimization problem of Subproblem 3 can be developed as:(5)$$\begin{array}{cc} \; & \mathrm{Objective}\;\mathrm{function}\text{:}\;\\ \; & \min\;\sum_{i\in I}\sum_{\ell \in T_{i}}\sum_{s\in S}(-\mu_{i}^{1}+\mu_{s}^{2}+\mu_{s}^{3}D_{i\ell }+\mu_{s}^{4}G_{i\ell }+\mu_{s}^{5}Q_{i\ell }\\ \; & \;\;\;\;\;\;\;\;\;\;\;\;\;\;\;\;\;+\mu_{i\ell s}^{9}+\sum_{j\in W_{i}}\mu_{ij\ell s}^{10})b_{i\ell s}\\ \; & \mathrm{subject}\;\mathrm{to}\;C3\text{:}\;\sum_{s\in S}b_{i\ell s}\leq 1,\;\forall i\in I,\;\forall \ell \in T_{i},\\ \; & \;\;\;\;\;\;\;\;\;\;\;\;\;\;\;\;\;b_{i\ell s}\in \left\{0,1\right\}. \end{array}$$

The solution process of Equation (Equation 5) is similar to that of Equation (Equation 4) and can be also divided into $\left|I|\right|T_{i}\left||S\right|$ subproblems. The decision variable $b_{i\ell s}$ is set to one when the minimum coefficient $\left(-\mu_{i}^{1}+\mu_{s}^{2}+\mu_{s}^{3}D_{i\ell }+\mu_{s}^{4}G_{i\ell }+\mu_{s}^{5}Q_{i\ell }+\mu_{i\ell s}^{9}+\sum_{j\in W_{i}}\mu_{ij\ell s}^{10}\right)$ is less than zero and corresponds to alliance subindices *i*, *ℓ*, and *s*. Otherwise, $b_{i\ell s}$ is set to zero. The run time of this subproblem is $O(|I||T_{i}||S||W_{i}|)$. The pseudocode is illustrated in Algorithm 3.
**Algorithm 3:** Subproblem 3.**Input:** Given parameters D,G,Q and Lagrangian    multipliers $\mu^{1},\mu^{2},\mu^{3},\mu^{4},\mu^{5},\mu^{9},\mu^{10}$.**Output:** Decision variable b.**Initialize:**$b_{i\ell s}\leftarrow 0$, $\forall i\in I,\forall \ell \in T_{i},\forall s\in S$**for**i=0 to (|I|−1)
**do**    **for**
ℓ=0 to $\left(|T_{i}|-1\right)$
**do**        **for**
s=0 to (|S|−1)
**do**           $c_{s}\leftarrow -\mu_{i}^{1}+\mu_{s}^{2}+\mu_{s}^{3}D_{i\ell }+\mu_{s}^{4}G_{i\ell }+\mu_{s}^{5}Q_{i\ell }$              $+\mu_{i\ell s}^{9}+\sum_{j\in W_{i}}\mu_{ij\ell s}^{10}$        **end for**        Find the index *m* that $c_{m}$ has the minimum value        in c.        **if**
$c_{m}<0$
**then**           $b_{i\ell m}\leftarrow 1$        **end if**    **end for****end for**

#### 4.2.4. Subproblem 4 (Related to $f_{ij\ell s}$)

By extracting items with decision variable $f_{ij\ell s}$, the optimization problem of Subproblem 4 can be developed as:(6)$$\begin{array}{cc} \; & \mathrm{Objective}\;\mathrm{function}\text{:}\;\\ \; & \min\;\sum_{i\in I}\sum_{j\in W_{i}}\sum_{\ell \in T_{i}}\sum_{s\in S}\left(\mu_{s}^{6}X_{ij\ell }-\mu_{s}^{7}X_{ij\ell }-\mu_{ij\ell s}^{10}\right)f_{ij\ell s}\\ \; & \mathrm{subject}\;\mathrm{to}\text{:}\;f_{ij\ell s}\in \left\{0,1\right\}. \end{array}$$

Equation (Equation 6) can be divided into $\left|I|\right|W_{i}\left||S|\right|T_{i}|$ cases. For the alliance subindices *i*, *j*, *ℓ*, and *s*, the decision variable $f_{ij\ell s}$ is set to one when the coefficient $\left(\mu_{s}^{6}X_{ij\ell }-\mu_{s}^{7}X_{ij\ell }-\mu_{ij\ell s}^{10}\right)$ is less than zero. Otherwise, $f_{ij\ell s}$ is set to zero. The run time is $O(|I||W_{i}||S||T_{i}|)$. The pseudocode is illustrated in Algorithm 4.
**Algorithm 4:**Subproblem 4.**Input:** Given parameters X and Lagrangian multipliers    $\mu^{6}$, $\mu^{7}$, $\mu^{10}$.**Output:** Decision variable f.**Initialize:**$f_{ij\ell s}\leftarrow 0$, $\forall i\in I,\forall j\in W_{i},\forall \ell \in T_{i},\forall s\in S$**for**i=0 to (|I|−1)
**do**    **for**
j=0 to (|Wi|−1)
**do**        **for**
ℓ=0 to $\left(|T_{i}|-1\right)$
**do**           **for**
s=0 to (|S|−1)
**do**               $c\leftarrow \mu_{s}^{6}X_{ij\ell }-\mu_{s}^{7}X_{ij\ell }-\mu_{ij\ell s}^{10}$               **if**
c<0
**then**                   $f_{ij\ell s}\leftarrow 1$               **end if**           **end for**        **end for**    **end for****end for**

#### 4.2.5. Subproblem 5 (Related to $z_{s}$)

By extracting items with decision variable $z_{s}$, the optimization problem of Subproblem 5 can be developed as:(7)$$\begin{array}{cc} \; & \mathrm{Objective}\;\mathrm{function}\text{:}\;\\ \; & \min\;\sum_{s\in S}\left[\beta_{s}-\sum_{i\in I}\left(\sum_{j\in W_{i}}\mu_{ijs}^{8}+\sum_{\ell \in T_{i}}\mu_{i\ell s}^{9}\right)\right]z_{s}\\ \; & \mathrm{subject}\;\mathrm{to}\text{:}\;z_{s}\in \left\{0,1\right\}. \end{array}$$

Equation (Equation 7) can be divided into $\left|S||I|\right|W_{i}|$ or $\left|S||I|\right|T_{i}|$ cases. In each case with subindex *s*, the decision variable $z_{s}$ is set to one when the coefficient $\left[\beta_{s}-\sum_{i\in I}\left(\sum_{j\in W_{i}}\mu_{ijs}^{8}+\sum_{\ell \in T_{i}}\mu_{i\ell s}^{9}\right)\right]$, which corresponds to alliance subindex *s*, is less than zero. Otherwise, $z_{s}$ is set to zero. The run time is $O(|S||I||W_{i}|)$ or $O(|S||I||T_{i}|)$, and the pseudocode is illustrated in Algorithm 5.
**Algorithm 5:** Subproblem 5.**Input:** Given parameters β and Lagrangian multipliers    $\mu^{8}$, $\mu^{9}$.**Output:** Decision variable z.**Initialize:**$z_{s}\leftarrow 0$, ∀s∈S**for**s=0 to (|S|−1)
**do**    c←0    **for**
i=0 to (|I|−1)
**do**        **for**
j=0 to $\left(|W_{i}|-1\right)$
**do**           $c\leftarrow c+\mu_{ijs}^{8}$        **end for**        **for**
ℓ=0 to $\left(|T_{i}|-1\right)$
**do**           $c\leftarrow c+\mu_{i\ell s}^{9}$        **end for**    **end for**    **if**
$\beta_{s}-c<0$
**then**        $z_{s}\leftarrow 1$    **end if****end for**

### 4.3. Procedure of Step 4: Dual Problem and the Subgradient Method

According to the weak Lagrangian duality theorem [30], the objective values of the LR problem $Z_{\mathrm{LR}}$ are the lower bounds (LBs) of the primal problem $Z_{\mathrm{IP}}$ with multiples $\mu_{i}^{1},\mu_{s}^{2},\mu_{s}^{3},\mu_{s}^{4},\mu_{s}^{5},\mu_{s}^{6},\mu_{s}^{7},\mu_{ijs}^{8},\mu_{i\ell s}^{9},\mu_{ij\ell s}^{10}\geq 0$, $\forall i\in I,\;\forall j\in W_{i},\;\forall \ell \in T_{i},\;\forall s\in S$. The formulation of the dual problem (D) is constructed to calculate the tightest LB ($\max Z_{D}$), where max $Z_{D}=\min Z_{\mathrm{LR}}$. Then, the dual problem can be formulated as: $$\begin{array}{cc} \; & \mathrm{Objective}\;\mathrm{function}\text{:}\;\\ \; & \max\;Z_{D}\\ \; & \mathrm{subject}\;\mathrm{to}\text{:}\;\mu^{1}\geq 0,\mu^{2}\geq 0,\mu^{3}\geq 0,\mu^{4}\geq 0,\mu^{5}\geq 0,\\ \; & \;\;\;\;\;\;\;\;\;\;\;\;\;\;\mu^{6}\geq 0,\mu^{7}\geq 0,\mu^{8}\geq 0,\mu^{9}\geq 0,\mu^{10}\geq 0. \end{array}$$

The subgradient method is commonly used for solving the dual problem by iteratively updating the Lagrangian multipliers [26,31,32].

First, let vector S be a subgradient of $Z_{D}$. In the qth iteration of the subgradient optimization procedure, the multiplier vector $\pi^{q}=\left(\mu^{1,q},\mu^{2,q},\mu^{3,q},\mu^{4,q},\mu^{5,q},\mu^{6,q},\mu^{7,q},\mu^{8,q},\mu^{9,q},\mu^{10,q}\right)$ is updated by $\pi^{q+1}=\pi^{q}+t^{q}S^{q}$ . The step size $t^{q}$ is determined by equation $t^{q}=\lambda \frac{\left(Z_{\mathrm{IP}}^{h}-Z_{D}\left(\pi^{q}\right)\right)}{∥S^{q}{∥}^{2}}$. The denominator $S^{q}$ is the sum of relaxed constraints concerning to the decision variable values based on the qth iteration. $Z_{\mathrm{IP}}^{h}$ is the primal objective value in the hth iteration. $Z_{D}\left(\pi^{q}\right)$ is the objective value of the dual problem in the qth iteration. λ is a constant, where 0≤λ≤2. Accordingly, the optimal objective value of the dual problem is obtained iteratively.

### 4.4. Procedure of Step 5: Obtaining the Primal Feasible Solutions

Applying the LR and the subgradient methods to solve the LR and dual problems determines a theoretical LB from the primal feasible solution. Crucial information regarding the primal feasible solution can be identified [28]. The feasible region of a mathematical programming problem defined by the solutions must be satisfied by all constraints. A set of primal feasible solutions to $Z_{\mathrm{IP}}$ is a subset of the infeasible region solutions to $Z_{\mathrm{LR}}$ or $Z_{D}$. Several alternative methods can be used sophisticatedly to obtain the primal feasible solutions from the observations of the infeasible region. For example, Figure 4 presents an experimental case. The green curve represents the process of obtaining the primal feasible solutions iteratively. The objective is to identify the minimum value of the primal problem ($Z_{\mathrm{IP}}^{h}$). Then, the LBs are determined using the subgradient method to iteratively obtain the tightest LB (max $Z_{D}$), represented by the purple line. The proposed resource management approach for obtaining primal feasible solutions is called the drop-and-add algorithm. It is a heuristic-based allocation mechanism for optimizing the objective function. It searches for a solution that satisfies not only all the user demands, but also the constraints. The initial solution is adopted using next-fit (NF) or first-fit (FF) as baselines for evaluating solution quality [33]. The proposed algorithm, FF, and NF are simultaneously implemented for result comparison.

### 4.5. Procedure of Step 6: Drop-and-Add Algorithm

Lagrangian multipliers determined from the dual problem have significant values for evaluating the sensitivity of objective value improvement [26,28,30,32]. Through the subproblems, the integrated weighting factor of applications is represented in (Equation 3) as $\left(-R_{i}-P_{i}+\mu_{i}^{1}|W_{i}|+\mu_{i}^{1}|T_{i}|\right)$; therefore, the sum of the values can be used as an index to interpret the application’s significance of *i*. The corresponding multipliers determine the ordering set of admitting, assigning, and scheduling decision variables. The other decision variables $a_{ijs}$ or $b_{i\ell s}$ and the assignment of indices *j*, *ℓ*, and *s* can be regarded as two bin-packing problems. The VM *j* can be packed into the server *s*. The VM *ℓ* can be packed into servers except server *s* to comply with the HA conditions for $b_{i\ell s}$. For performance evaluation, NF is adopted for sequentially performing the algorithm of assignment for $a_{ijs}$ and then $b_{i\ell s}$. The flowchart of the drop-and-add algorithm is shown in Figure 5.

## 5. Computational Experiments

A cloud service provider’s resource parameters and the applications’ demand attributes were simulated in a cloud computing experimental environment and are presented in Table 6. U(x,y) means a number uniform distributed between the parameters of *x* and *y*.

The algorithms were constructed and implemented to analyze solution quality and improvement ratios (IRs) in several simulation cases. Solution quality is defined as the objective value gap (denoted as GAP) between the proposed algorithm and the LR problem, which is expressed as $\mathrm{GAP}=\frac{|V_{\mathrm{Drop\&Add}}-V_{\mathrm{LR}}|}{\left|\max(V_{\mathrm{Drop\&Add}},V_{\mathrm{LR}})\right|}\times 100\%$, where $V_{\mathrm{Drop\&Add}}$ is the objective value of applying the drop-and-add algorithm and $V_{\mathrm{LR}}$ is the objective value of the LR problem. ${\mathrm{IR}}^{\mathrm{NF}}$ is expressed as ${\mathrm{IR}}^{\mathrm{NF}}=\frac{V_{\mathrm{Drop\&Add}}-V_{\mathrm{NextFit}}}{\left|\max(V_{\mathrm{NextFit}},V_{\mathrm{Drop\&Add}})\right|}\times 100\%$. ${\mathrm{IR}}^{\mathrm{FF}}$ is expressed as ${\mathrm{IR}}^{\mathrm{FF}}=\frac{V_{\mathrm{Drop\&Add}}-V_{\mathrm{FirstFit}}}{\left|\max(V_{\mathrm{FirstFit}},V_{\mathrm{Drop\&Add}})\right|}\times 100\%$, where $V_{\mathrm{NextFit}}$ and $V_{\mathrm{FirstFit}}$ are the objective values of employing the NF algorithm or FF algorithm, respectively. The experiments were developed in Python and implemented in a VM on a workstation with a quad-core CPU, 8 GB RAM, and Ubuntu 14.04. The traffic loads of tasks and arrival time intervals were randomly generated. The experimental environment was initialized for a data center including edge and core clouds. VMs were requested by applications to represent computing requirements. Based on resource admission control strategies and scheduling models, the VMs were packed into the corresponding servers in the cloud computing environments.

### 5.1. Performance Evaluation Case: Traffic Load

This experiment was designed to analyze VMs requested by applications in a time slot, which can be interpreted as a snapshot of the system loading. First, the trend of the objective function with the number of applications as the control variable was examined. The result in Figure 6 indicates that the objective value of (IP)increases with the number of applications. The drop-and-add and LB values were almost the same in some cases (20∼80), indicating that the optimal solution was determined when the GAP was less than 0.80% (the minimized one was 0.60%). Table 7 compares drop-and-add with NF or FF (higher values are preferable) in the primal problem. The maximum IR was 89.48%, with 180 applications in both FF and NF. The penalty of unsatisfied applications significantly increased the objective value for indicating when arriving applications exceeded system capacity with the NF or FF algorithm in the cases of the number of applications being over 100. Otherwise, the drop-and-add algorithm can select valuable applications to pack into servers, which results in a higher objective value than NF or FF in the cases of the number of applications being over 100.

### 5.2. Performance Evaluation Case: Number of Servers

The cost function is emulated as the capital expenditure (CAPEX) of the cloudlets and core clouds related to network infrastructure deployment of appropriate levels of servers, where the budget is a significant constraint. The cost difference between high- and low-end servers is also typically significant. Service providers generally conduct a comprehensive system analysis to determine the methods of how to manage, control, and operate a system appropriately. The plans, such as turning servers on or off, provide sufficient QoS to applications efficiently and effectively. Furthermore, the servers purchased were deployed at four levels with different capacities in this experiment. Nonhomogeneous servers were deployed under the same limited budget. The rapid increases in data traffic are represented as traffic loads to determine which method delivered superior QoS for applications at affordable costs and preserved revenue. The following are the experimental scenarios tested. Figure 7 and Table 8 present the results of the proposed methods (drop-and-add, LB, NF, and FF). The drop-and-add algorithm attained the most practical objective value (higher is preferable) compared with NF and FF in the case of 40.

Regarding the GAP, some values in the other situations were calculated to determine the difference in rate value between drop-and-add and LB. The GAP values indicate that the minimum value (44.62%) was determined in one of the cases with numerous servers (40). A data center has sufficient resource space in servers with the drop-and-add, NF, and FF algorithms. The maximum improvement ratio is represented by the ratio of improvement of feasible solutions and was 193.73% in numerous servers (i.e., 40). The result reveals that the resource allocation algorithm has a significant impact on system performance with a limited resource constraint.

### 5.3. Performance Evaluation Case: Effect of HA

As far as crucial business requirements are concerned, cloud service providers should offer high quality, excellent availability, good performance, and reliable services. In this case, the applications are divided into levels by using VM replication for HA requests. The VMs requesting to be in the $T_{i}$ set with HA by the application *i* asking for VM placement must be mutually and exclusively allocated into different physical servers. The level of exclusivity is called the dual-rate. In the subsequent experiment, the dual-rate was the parameter configured for HA VMs. Moreover, a dual-rate of 0.5 indicates that the HA VMs, $|T_{i}|$, requested by half of $|W_{i}|$ for application *i* are allocated to different servers. The dual-rate equals 0.3, which indicates 30% of standard VMs required for application *i* with HA capability. In Figure 8, it is evident that the dual-rate significantly affected the application satisfaction rate. Thus, the drop-and-add algorithm offers more benefits (higher values are preferable) than NF or FF in the dual-rate cases. The improvement ratios increased significantly when the dual-rate was higher than 0.3, and the maximum IR was 608.39%. The drop-and-add algorithm achieved flexibility and efficiency and could sufficiently obtain the maximum objective value to deal with HA requests. As observed in Table 9, FF and NF performed poorly when the dual-rate was beyond 0.3. The tasks were not assigned to appropriate servers with HA. This resulted in turning on more servers, which corresponded to cost generation.

### 5.4. Performance Evaluation Case: Time complexity comparison

Table 10 shows the time complexity of the LR-based solution for resource management in a sliced network. We added an existing scheme, brute force, for comparison in Table 10. The time complexity of this scheme was higher than the proposed LR-based algorithm. Furthermore, the corresponding explanations were included to support our statements. The time complexity of the proposed LR-based solutions was $O(N|I||W_{i}||T_{i}||S|)$. Table 10 shows the time complexity of the LR-based solution. The Lagrange dual solution was determined by Sub-problems (Equation 4)–(Equation 6), which were solved using the minimum algorithm with |S| servers. Each sub-problem required O(|I||Wi||Ti|) time minimum coefficient among servers. Since the sub-problems were solved individually by divide-and-conquer algorithms, the time complexity for each of the sub-problems was constant. Thus, the worst case of time complexity among these subproblems was considered significant in each iteration. The Lagrange dual solutions for the sub-problems could be obtained after a maximum number of iterations *N*, and the time complexity was $O(N|I||W_{i}||T_{i}||S|)$. The number of iterations pre-defined to converge was about 600 based on the experiment results shown in Figure 4. Fortunately, all given parameters and multipliers for the solution did not need the same initial values. The convergence was achieved in a small number of iterations with the previous execution results. Furthermore, the proposed algorithms can be set as the output results at any complete iteration. Thus, the time can be controlled in practice.

### 5.5. Discussion

To test the proposed algorithm, we designed three experiments involving changes in application demands, the cost of servers, and dual-rates for HA with application requests, as shown in Table 11. The research problem was formulated as a mathematical programming problem to determine both admission control and resource scheduling problems from a service provider perspective. Overall, the drop-and-add algorithm is the most effective for obtaining the optimal solution in the fewest iterations. The optimization-based energy-efficient admission control and resource allocation algorithms have significant benefits in cloud computing systems. The significance of applications was inspired by the coefficient of (Equation 3), which sorts applications by a composition of application weights, client satisfaction, and high availability. Therefore, using the LR-based solution approach with multipliers to obtain the primal feasible solutions was suitable for allocating VMs to servers efficiently and effectively. The mathematical formulation was decomposed into five subproblems. It was solved optimally using parallel computation techniques to reduce computation time substantially. Furthermore, the experimental results reveal and confirm that the drop-and-add algorithm was active in a few minutes by the most significant in the sliced network stages shown in Table 7, Table 8, Table 9, Table 10 and Table 11. The following suggested problems and limitations of this paper can be further studied and solved: QoS requirements can be classified into more categories. Resource sharing between multiple service providers is a new research problem that necessitates considering related issues, such as the sharing economy, from multiple network service provider perspectives. Hence, operator pricing policies could be a worthwhile topic.

## 6. Summary and Conclusions

A mathematical programming model is used to develop resource management by simulating the cloud service provider role in this paper’s cloud computing systems in 5G network slices. The mathematical model is solved using an LR-based approach. The proposed algorithm increases the cloud computing network infrastructure’s flexibility, including cloudlets, and core clouds, to maximize rewards by admitting as many applications as possible. The gaps between upper bounds and lower bounds in the computational experiments demonstrate the drop-and-add heuristic optimal solution qualities. The main contribution is demonstrating that the orchestrator designed the resource management algorithm significantly determined using Lagrangian multipliers to indicate task significance. A promising and straightforward resource allocation approach is proposed to combine client satisfaction and high availability for network planning in a sliced network. The developed optimization-based efficient admission control and resource allocation algorithms are confirmed through various experimental cases. The proposed method has excellent effectiveness and efficiency compared with the LB, FF, and NF solutions based on the experimental results. The resource management mechanisms enables the slicing network as services to efficiently and maximize system revenue in 5G networks.

## Figures and Tables

**Figure 1 sensors-21-01882-f001:**
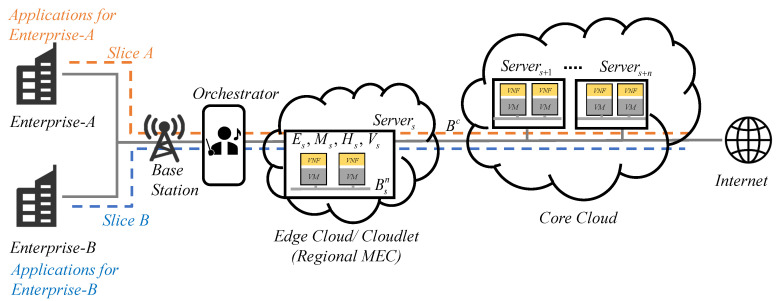
Network topology in slicing networks.

**Figure 2 sensors-21-01882-f002:**
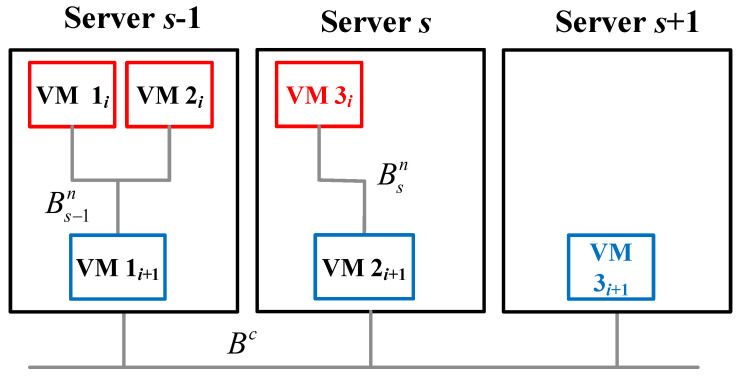
Representation of internal and external transmissions between VMs in servers.

**Figure 3 sensors-21-01882-f003:**
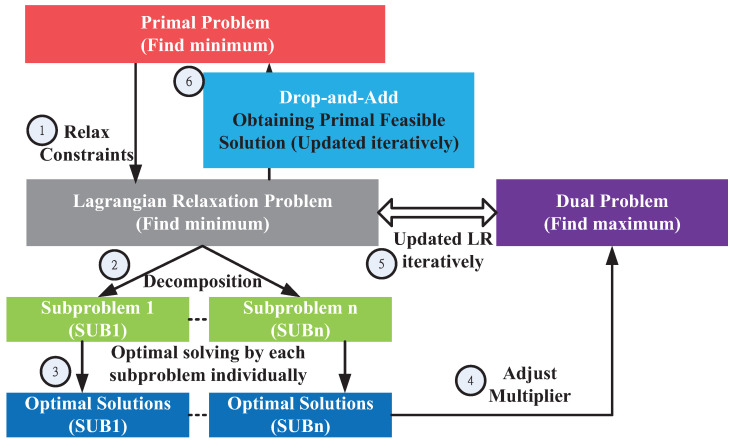
Lagrangian relaxation-based solution process flow.

**Figure 4 sensors-21-01882-f004:**
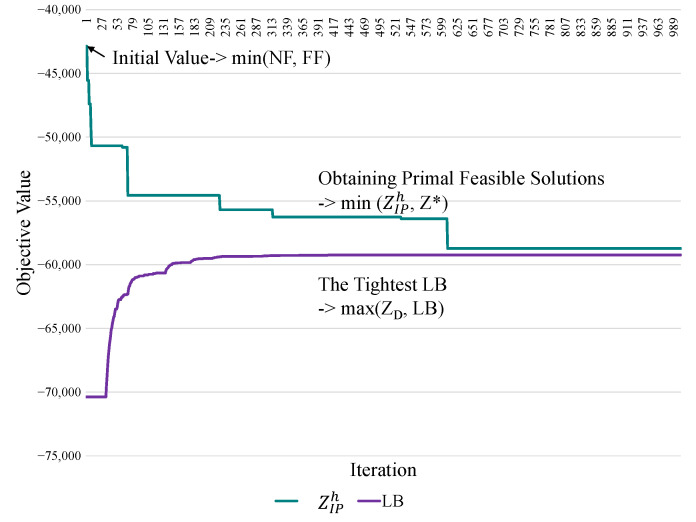
Obtaining primal feasible solutions and the tightest lower bound (LB). NF, next-fit.

**Figure 5 sensors-21-01882-f005:**
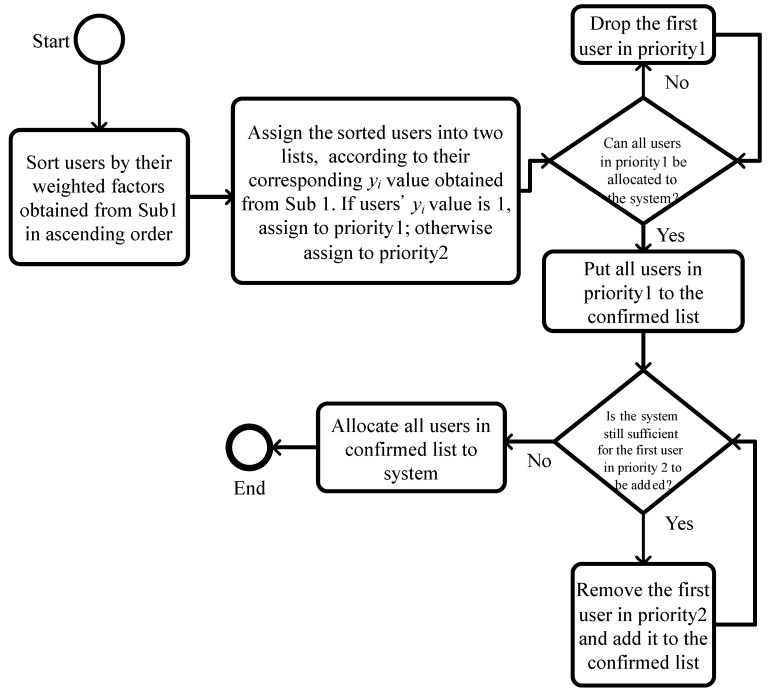
Flowchart of the drop-and-add algorithm.

**Figure 6 sensors-21-01882-f006:**
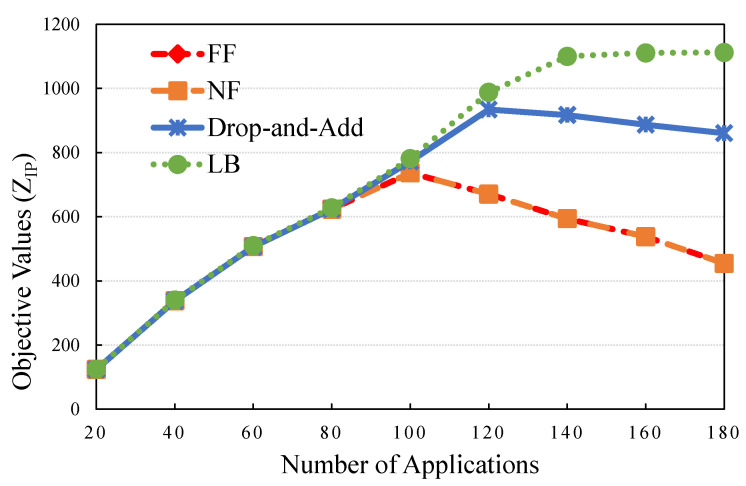
Objective value by the number of applications.

**Figure 7 sensors-21-01882-f007:**
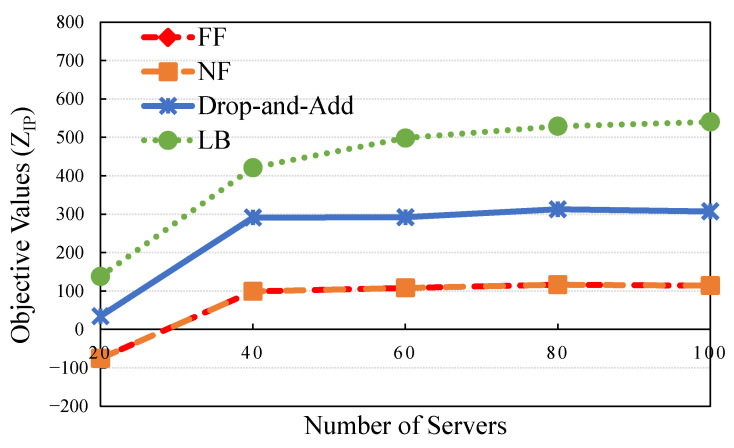
Objective value by the number of servers.

**Figure 8 sensors-21-01882-f008:**
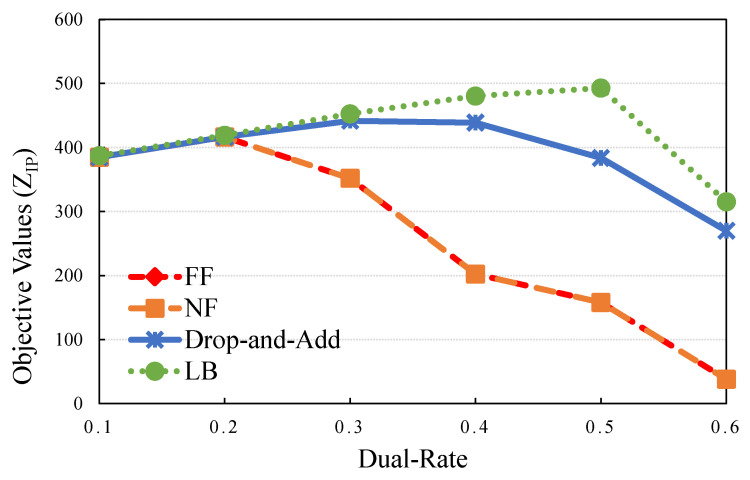
Objective value when scaling the dual-rate.

**Table 4 sensors-21-01882-t004:** Given parameters.

Notation	Description
*S*	Index set of physical servers in the cloud computing system (cloudlets and core clouds), where S={1,2,…,s,…,|S|}.
*I*	Index set of applications on the cloud computing system, where I={1,2,…,i,…,|I|}.
*W*	Index set of standard VMs, $W={⋃}_{i=1}^{I}W_{i}$ and $W_{i}=\left\{1,2,\ldots ,j,\ldots ,\right|W_{i}\left|\right\}$, where $W_{i}$ is also an index set of standard VMs required by application *i*, where i∈I.
*T*	Index set of VMs with high availability (HA), $T={⋃}_{i=1}^{I}T_{i}$ and $T_{i}=\left\{1,2,\ldots ,\ell ,\ldots ,\right|T_{i}\left|\right\}$, where $T_{i}$ is also an index set of VMs with HA required by application *i*, where i∈I.
*N*	Set of total VMs, $N={⋃}_{i=1}^{I}N_{i}$ and $N_{i}=W_{i}⋃T_{i}$, where $N_{i}$ is the total number of standard VMs and VMs with HA required by application *i*, where i∈I.
*r*	Dual-rate; represents the ratio between the number of standard VMs and the VMs with HA, formulated as $T_{i}=\left\lceil \left|W_{i}\right|\times r\right\rceil$, where i∈I.
$B{}_{s}^{n}$	Internal transmission bandwidth of server *s*, where s∈S.
$B{}^{c}$	Shared transmission bandwidth within the cloud computing system (cloudlets and core clouds).
$E_{s}$	Processing capability of each central processing unit (CPU) core in server *s*, where s∈S.
$M_{s}$	Total random access memory (RAM) capacity in server *s*, where s∈S.
$H_{s}$	Total storage capacity in server *s*, where s∈S.
$V_{s}$	Maximum number of VMs allowable on server *s*, where s∈S.
$\beta_{s}$	Cost rate for opening server *s*, where s∈S.
$R_{i}$	Reward of admitting application *i* (application *i* can be admitted only if the demands on all types of resources are fully satisfied), where i∈I.
$P_{i}$	Penalty of rejecting application *i* (application *i* is rejected only if all of the types of resource requirements are not fully satisfied), where i∈I.
$D_{ij}$	Total CPU processing capability on VM *j* required by application *i*, where i∈I, $j\in W_{i}$.
$G_{ij}$	Total RAM capability required by application *i* on VM *j*, where i∈I, $j\in W_{i}$.
$Q_{ij}$	Total storage capability required by application *i* on VM *j*, where i∈I, $j\in W_{i}$.
$X_{ij\ell }$	Total transmission channel capacity required by application *i* between VMs *j* and *ℓ*, where i∈I, $j\in W_{i}$, $\ell \in T_{i}$.

**Table 5 sensors-21-01882-t005:** Decision variables.

Notation	Description
$y_{i}$	Binary variable, 1 if the application *i* is completely allocated to and served in the computing system, and 0 otherwise, where i∈I.
$a_{ijs}$	Binary variable used for task admission control and assignment, 1 if the standard VM *j* of application *i* is admitted in the cloud computing networks and allocated to server *s*, and 0 otherwise, where i∈I, $j\in W_{i}$, s∈S.
$b_{i\ell s}$	Binary variable used for task admission control and assignment, 1 if the VM with HA *ℓ* of application *i* is admitted in the cloud computing networks and allocated to server *s*, and 0 otherwise, where i∈I, $\ell \in T_{i}$, s∈S.
$f_{ij\ell s}$	Binary variable used for exclusive setting, 1 if VMs *j* and *ℓ* of application *i* are allocated to server *s*, and 0 otherwise, where i∈I, $j\in W_{i}$, $\ell \in T_{i}$, s∈S.
$z_{s}$	Binary variable for server power-on or power-off status, 1 if server *s* is turned on, and 0 otherwise, where s∈S.

**Table 6 sensors-21-01882-t006:** Given parameters for the experiments.

Given Parameter	Value
Number of servers, |S|	56–84
Number of applications, |I|	24–52
Dual-rate, *r*	0.5
Number of standard VMs for application *i*, $|W_{i}|$	$|W_{i}|∼U\left(1,10\right),\;\forall i\in I$
Number of VMs with HA for application *i*, $|T_{i}|$	$T_{i}=\left\lceil \left|W_{i}\right|\times r\right\rceil ,\forall i\in I$
Total number of VMs for application *i*, $|N_{i}|$	$|N_{i}\left|=\right|W_{i}\left|+\right|T_{i}|,\;\forall i\in I$
Host internal bandwidth capacity, $B_{s}^{n}$ (Mbps)	120–225
Shared transmission bandwidth within the cloud computing system, $B^{c}$ (Mbps)	1000
Host CPU processing capacity, $E_{s}$ (GHz)	480–900
Host memory capacity, $M_{s}$ (GB)	120–225
Host storage capacity, $H_{s}$ (GB)	1200–2250
The maximum number of VMs allowable on server *s*, $V_{s}$	8–15
Cost rate for opening server *s*, $\beta_{s}$	$\beta_{s}=\left(E_{s}+M_{s}+H_{s}+V_{s}\right)/$40,000
Reward rate of each application, $R_{i}$	$R_{i}∼\sum_{j}\left(\frac{D_{ij}}{50}\right),\;\forall i\in I$
Penalty rate of each application, $P_{i}$	$P_{i}∼\sum_{j}\left(\frac{D_{ij}}{100}\right),\;\forall i\in I$
CPU requests of a task, $D_{ij}$ (GHz)	$D_{ij}∼U\left(1,120\right),\;\forall i\in I,\;\forall j\in W_{i}$
Memory requests of a task, $G_{ij}$ (GB)	$G_{ij}∼U\left(1,30\right),\;\forall i\in I,\;\forall j\in W_{i}$
Storage requests of a task, $Q_{ij}$ (TB)	$Q_{ij}∼U\left(1,300\right),\;\forall i\in I,\;\forall j\in W_{i}$
Total transmission channel capacity, $X_{ij\ell }$ (Mbps)	$X_{ij\ell }∼U\left(0,30\right),\;\forall i\in I,\;\forall j\in W_{i},\;\forall \ell \in T_{i}$
Bandwidth requests of a task, $C_{ijs}$ (Mbps)	$C_{ijs}∼U\left(1,2000\right),\;\forall i\in I,\;\forall j\in W_{i},s\;\forall s\in S$

**Table 7 sensors-21-01882-t007:** Comparison of solution qualities under different number of applications.

Number of Applications	Drop-And-Add	LB	Gap (%)	NF	FF	${\mathrm{IR}}^{\mathrm{NF}}$ (%)	${\mathrm{IR}}^{\mathrm{FF}}$ (%)
20	123.743	124.760	0.80	123.687	123.687	0.05	0.05
40	337.910	340.080	0.64	337.910	337.910	0.00	0.00
60	506.729	509.780	0.60	506.729	506.729	0.00	0.00
80	623.692	627.760	0.65	623.523	623.523	0.02	0.02
100	772.445	780.680	1.06	736.595	736.595	4.87	4.87
120	934.295	987.845	5.73	670.535	670.535	39.33	39.33
140	916.985	1099.91	19.94	593.945	593.945	54.39	54.39
160	886.925	1110.70	25.23	537.905	537.905	64.89	64.89
180	860.705	1112.32	29.23	454.235	454.235	89.48	89.48

In some cases (20∼80), the lower bound (LB) value is extremely close to, but not equal to that of the drop-and-add algorithm. In other cases, the drop-and-add algorithm has a significant improvement rate compared with NF or FF.

**Table 8 sensors-21-01882-t008:** Comparison of solution qualities under different number of servers.

Number of Servers	Drop-And-Add	LB	Gap (%)	NF	FF	${\mathrm{IR}}^{\mathrm{NF}}$ (%)	${\mathrm{IR}}^{\mathrm{FF}}$ (%)
20	34.439	137.926	300.5	−75.601	−75.601	145.55	145.55
40	291.198	421.117	44.62	99.138	99.138	193.73	193.73
60	291.880	498.201	70.69	108.01	108.01	170.23	170.23
80	312.825	528.898	69.07	116.41	116.41	168.73	168.73
100	307.056	540.088	75.89	113.894	113.894	169.6	169.6

**Table 9 sensors-21-01882-t009:** Comparison of solution qualities under different the dual-rate.

Dual-Rate	Drop-and-Add	LB	Gap (%)	NF	FF	${\mathrm{IR}}^{\mathrm{NF}}$ (%)	${\mathrm{IR}}^{\mathrm{FF}}$ (%)
0.1	384.827	387.200	0.61	384.827	384.827	0.00	0.00
0.2	416.531	419.040	0.60	416.531	416.531	0.00	0.00
0.3	441.730	452.618	2.46	351.926	351.926	25.52	25.52
0.4	438.900	480.493	9.48	202.224	202.224	117.04	117.04
0.5	383.708	492.960	28.47	158.026	158.026	142.81	142.81
0.6	269.938	315.366	16.83	38.106	38.106	608.39	608.39

**Table 10 sensors-21-01882-t010:** Time complexity comparison between the proposed LR-based approach and other schemes.

Algorithm	Time Complexity	Annotation
Equation (Equation 3)	O(|I|)	|I| subproblems determine the value of the decision variable for each application *i*.
Equation (Equation 4)	$O(|I||W_{i}||S||T_{i}|)$	$\left|I|\right|W_{i}|$ subproblems with summation of $\ell \in T_{i}$, where i∈I, determine the binary decision variable $a_{ijs}$ that the coefficient is minimized among |S|.
Equation (Equation 5)	$O(|I||T_{i}||S||W_{i}|)$	$\left|I|\right|T_{i}|$ subproblems with summation of $j\in W_{i}$, where i∈I determine the binary decision variable $b_{i\ell s}$ that the coefficient is minimized among |S|.
Equation (Equation 6)	$O(|I||W_{i}||S||T_{i}|)$	$\left|I|\right|W_{i}\left||S|\right|T_{i}|$ subproblems determine decision variable $f_{ij\ell s}$.
Equation (Equation 7)	$O(|S||I||W_{i}|)$ or $O(|S||I||T_{i}|)$	|S| subproblems with summation $\mu_{ijs}^{8}$ or $\mu_{i\ell s}^{9}$ determine decision variable $z_{s}$.
Dual problem (D)	$O(N|I||W_{i}||T_{i}||S|)$	*N* times of the maximum complexity by the sub-problems, (Equation 4)–(Equation 6).
Drop-and-add algorithm	$O(N|I||W_{i}||T_{i}||S|)$	The algorithm adjusts the values of decision variables $y_{i}$, $a_{ijs}$, $b_{i\ell s}$, $f_{ij\ell s}$, and $z_{s}$ based on the dual problems. The complexity is the worst case determined by the decision variables of the subproblems.
FF	$O(|I||W_{i}||T_{i}||S|)$	The resource allocation.
NF	$O(|I||W_{i}||T_{i}||S|)$	The resource allocation.
Brute force	$O(2^{{\left|I\right|}^{4}|W_{i}{|}^{2}|T_{i}{|}^{2}{\left|S\right|}^{4}})$	The total combination of decision variables.

**Table 11 sensors-21-01882-t011:** Execution time in the experiments.

Traffic load (number of servers: 80, dual-rate: 0.5)			
**Number of applications**	**60**	**120**	**180**
min (FF, NF)	0.0418 s.	0.0826 s.	0.1067 s.
Proposed method	373 s.	744 s.	1117 s.
Number of servers (number of users: 80, dual-rate: 0.5)			
**Number of servers**	**40**	**80**	**100**
min (FF, NF)	0.213 s.	0.159 s.	0.269 s.
Proposed method	236 s.	478 s.	604 s.
Effect of HA (number of applications: 80, number of servers: 80)			
**Dual-rate**	**0.2**	**0.4**	**0.6**
min (FF, NF)	0.059 s.	0.246 s.	0.430 s.
Proposed method	284 s.	437 s.	583 s.

## Data Availability

Not applicable.

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
