# Peer review of "Optimization-Based Resource Management Algorithms with Considerations of Client Satisfaction and High Availability in Elastic 5G Network Slices"

_sensors, 2021, doi:10.3390/s21051882_

Round 1

Reviewer 1 Report

This work proposes an orchestrator with a mathematical programming model in a global viewpoint to solve resource management problems with satisfying the clients’high availability requirements. I have the following questions:

(1)They should clearly state the innovations from the application point of view and they should clearly define which are the innovative features of their proposal with respect to adopted logics.

(2)The authors also should compare the computational time complexity of the proposed method with other new schemes. They should clearly state the point.

(3) It is generally known that the proposed algorithm is sensitive to the initial values. I want to know whether the initial value can influence the results or not? And how do you select the initial value?

(4)Is the complexity of the proposed scheme very high? Is it difficult to use in practice?

(5)The authors also should compare the computational time complexity of the proposed method with other new schemes. They should clearly state the point.

(6)The literature review of 5G still needs to be improved. For example:

Intelligent Outage Probability Prediction for Mobile IoT Networks Based on an IGWO-Elman Neural Network, IEEE Transactions on Vehicular Technology, 2021.DOI: 10.1109/TVT.2021.3051966.

I/Q Imbalance Aware Nonlinear Wireless-Powered Relaying of B5G Networks: Security and Reliability Analysis, IEEE Transactions on Network Science and Engineering, 2020. doi: 10.1109/TNSE.2020.3020950.

(7)The use of the present tense in Conclusions is not correct .

Reviewer 2 Report

In their resource allocation optimization, the authors do not consider the 5G modems' computation power, which is very high due to the very complex signal processing in the 5G network. 
The next critical resource that was not considered is the frequency spectrum. The management of the frequency spectrum is considerably different than the allocation of the computation resources or energy. We cannot simply add additional frequencies in contrast with energy or send the data to the other server.   
Authors should extend their analyses of the resources mentioned above.

Round 2

Reviewer 1 Report

It is revised.